# Enhanced Subcellular Trafficking of Resveratrol Using Mitochondriotropic Liposomes in Cancer Cells

**DOI:** 10.3390/pharmaceutics11080423

**Published:** 2019-08-20

**Authors:** Ji Hee Kang, Young Tag Ko

**Affiliations:** College of Pharmacy and Gachon Institute of Pharmaceutical Sciences, Gachon University, Incheon 21936, Korea

**Keywords:** mitochondrial targeting, triphenylphosphonium cation, dequalinium, liposomes, resveratrol

## Abstract

Mitochondria are membrane-enclosed organelles present in most eukaryotic cells, described as “power houses of the cell”. The mitochondria can be a target for inducing cancer cell death and for developing strategies to bypass multi drug resistance (MDR) mechanisms. 4-Carboxybutyl triphenylphosphonium bromide-polyethylene glycol-distearoylphosphatidylethanolamine (TPP-DSPE-PEG) and dequalinium-polyethylene glycol-distearoylphosphatidylethanolamine (DQA-DSPE-PEG) were synthesized as mitochondriotropic molecules. Mitochondria-targeting liposomes carrying resveratrol were constructed by modifying the liposome’s surface with TPP-PEG or DQA-PEG, resulting in TLS (Res) and DLS (Res), respectively, with the aim to obtain longer blood circulation and enhanced permeability and retention (EPR). Both TLS (Res) and DLS (Res) showed dimensions of approximately 120 nm and a slightly positive zeta potential. The enhanced cellular uptake and selective accumulation of TLS (Res) and DLS (Res) into the mitochondria were demonstrated by behavioral observation of rhodamine-labeled TLS or DLS, using confocal microscopy, and by resveratrol quantification in the intracellular organelle, using LC–MS/MS. Furthermore, TLS (Res) and DLS (Res) induced cytotoxicity of cancer cells by generating reactive oxygen species (ROS) and by dissipating the mitochondrial membrane potential. Our results demonstrated that TLS (Res) and DLS (Res) could provide a potential strategy to treat cancers by mitochondrial targeting delivery of therapeutics and stimulation of the mitochondrial signaling pathway.

## 1. Introduction

Mitochondria are membrane-enclosed organelles present in most eukaryotic cells, and are described as “power houses of the cell” [1,2]. The mitochondria supply the cells’ energy and play an important role in the regulation of the cell cycle, including cell growth, proliferation, differentiation and cell death [2,3]. Mitochondria in cancer cells are known to have structural and functional differences, such as increased and altered mitochondrial DNA, increased hexokinase production, and altered mitochondrial protein and lipid content, compared to that of normal cells [4,5,6,7,8]. Cancer cells also have twice the mitochondrial mass of normal cells, leading to more cellular ATP production and a greater mitochondrial potential [2,8,9].

The mitochondria can be a target for inducing cancer cell death and for developing a strategy to bypass multi drug resistance (MDR) [10]. Cancer cells need high energy levels for proliferation [11], thus, mitochondria-targeting drugs can lead to cancer cell death by extensive ATP depletion [12]. In addition, the strategy to block anti-apoptotic proteins in the mitochondria can result in the activation of the cell death machinery, composed of catabolic hydrolases and proteases [10,13]. Inducing apoptosis through direct targeting of cancer cell mitochondria could be a strategy to circumvent the MDR mechanism in chemotherapy [3,8,14]. Therefore, mitochondria-targeting chemotherapy is receiving much attention and recognition, as studies have shown that it plays a key role in apoptosis and necrosis and in the regulation of cancer [2,15,16].

Many mitochondrial drug delivery systems have been constructed with positively charged molecules, as the mitochondrial membranes of cancer cells are negatively charged [17,18]. Several studies have reported an improvement in the therapeutic efficacy of mitochondria-targeting agents, such as triphenylphosphonium cation [19], dequalinium (DQA) [2], rhodamine B [17] and mitochondria-targeting signal peptides (MTSs) [20].

4-Carboxybutyl triphenylphosphonium bromide (TPP) has a delocalized lipophilic cation as a structural feature, which can permeate mitochondrial membranes, leading to its accumulation or the facilitated access to the mitochondria [21]. The delocalized positive charge of TPP, which consists of three phenyl groups, facilitates their mitochondrial uptake driven by the highly negatively charged mitochondrial membrane of cancer cells [22].

DQA is a cationic amphiphile, which contains two cationic aminoquinaldinium rings, with delocalized charge centers [23]. This feature of DQA facilitates selective accumulation in the mitochondria of cancer cells, driven by the transmembrane electric potential [24].

Resveratrol is part of a group of compounds called polyphenols and is significantly present in grapes and red wine [25,26]. A huge amount of preclinical studies showed that resveratrol is a potential anticancer agent due to its chemopreventive ability in three major stages of carcinogenesis, including initiation, promotion and progression [2,27,28,29,30]. Resveratrol is able to induce cell death through the mitochondrial apoptotic pathway, in which the mitochondria play a central role in the release of pro-apoptotic factors [25,31]. However, its poor bioavailability, caused by its low water solubility, chemical instability and intestinal metabolism, are major obstacles for its use in clinical therapy [29,30,32].

The PEG-stabilized liposomes have been successfully applied for the in vivo delivery of various therapeutic agents, including nucleic acids, proteins and small molecules [2,33,34]. Liposomes consist of a phospholipid-formed bilayer and can be formulated in different sizes, ranging from a few to tens of micrometers [35,36]. The lower size range of liposomes has been heavily studied for the treatment of diseases because of their good ability to be taken up passively by cells and in tissues [34,37]. The liposomes are highly biocompatible, display low toxicity, can be loaded with both hydrophilic and hydrophobic drugs, and can be used as gene delivery vehicles [38,39]. An additional advantage of liposomes is that they can be modified for specific purposes through the modulation of their lipid composition or the addition of functional agents, like ligands for longer circulation, targeted and enhanced permeability and retention (EPR) [35,39,40,41].

In this study, we propose surface-modified liposomes with TPP or DQA for mitochondria-targeted delivery of resveratrol in cancer cells (Figure 1). To modify the surface of liposomes for mitochondrial targeting, 4-carboxybutyl triphenylphosphonium bromide- polyethylene glycol-distearoylphosphatidylethanolamine (TPP-DSPE-PEG) and dequalinium-polyethylene glycol-distearoylphosphatidylethanolamine (DQA-DSPE-PEG) were synthesized as mitochondriotropic molecules (Figure 2). Mitochondria-targeting liposomes carrying resveratrol were constructed by modifying the TPP-DSPE-PEG (TLS (Res)) or DQA-DSPE-PEG (DLS (Res)) on the surface of liposomes, with the aim to obtain longer blood circulation and an enhanced permeability and retention (EPR) effect. The objectives of the present study were to prepare the mitochondria-targeting liposomes carrying resveratrol, to define the action mechanisms, and to evaluate their efficacy in treating cancer cells.

## 2. Materials and Methods

### 2.1. Materials

1-Palmitoyl-2-oleoyl-*sn*-glycero-3-phosphocholine (POPC), 1,2-distearoyl-*sn*-glycero-3-phosphoethanol-amine-*N*-[methoxy(polyethylene glycol)-2000-COOH] (DSPE-PEG_2000_-COOH) and cholesterol were purchased from Avanti Polar Lipids (Alabaster, AL, USA). 2′,7′-Dichlorodihydrofluorescin diacetate (DCFH-DA) was obtained from Enzo Life Sciences (Farmingdale, NY, USA). Dequalinium (DQA) and 5,5′,6,6′-tetrachloro-1,1′,3,3′-tetraethylbenzimidazolocarbocyanine (JC-1) were obtained from Sigma-Aldrich (St. Louis, MO, USA), and resveratrol (Res) and (4-carboxybutyl) triphenylphosphonium bromide (TPP) were obtained from Tokyo Chemical Industry (Tokyo, Japan). All other reagents were purchased from Sigma-Aldrich or Tokyo Chemical Industry (Tokyo, Japan), unless otherwise stated.

### 2.2. Synthesis of TPP-DSPE-PEG and DQA-DSPE-PEG Conjugates

DSPE-PEG_2000_-COOH (25 mg, 0.0088 mmol) or DSPE-PEG_2000_-NH_2_ (25 mg, 0.0088 mmol) was allowed to react for 3 h with EDC-HCl (3.4 mg, 0.018 mmol) and NHS (1.01 mg, 0.0087 mmol) in 2 mL of chloroform in the presence of three drops of triethylamine at room temperature in the dark. The progress of the reaction was checked by thin-layer chromatography (TLC) on a silica gel plate. Next, dequalinium (DQA; 4.6 mg, 0.0087 mmol) or TPP (3.9 mg, 0.0087 mmol) dissolved in DMSO (2 mL) was added to DSPE-PEG2000-COO-NHS and stirred for 30 min. Chloroform was removed by an evaporator, and 5 mL of distilled water was added. The final mixture was dialyzed using a cellulose dialysis tubing (MWCO 3000) against deionized water for 48 h, and then freeze-dried using a lyophilizer. After freeze-drying, the residue was redissolved in distilled water and filtered through a 0.45-µm filter syringe. The liquid was then freeze-dried again and the formed product was characterized by proton NMR spectroscopy (Brucker, 600 MHz, Billerica, MA, USA) and MALDI-TOF mass spectrometer (AXIMA-Assurance, Shimadzu, Kyoto, Japan).

### 2.3. Preparation of TPP-DSPE-PEG-Modified Liposomes Carrying Resveratrol (TLS (Res)) and DQA-DSPE-PEG-Modified Liposomes Carrying Resveratrol (DLS (Res))

The TLS (Res) and DLS (Res) were prepared using a previously reported method with minor modifications [33]. The following lipids (total amount of 2 mg) were dissolved in chloroform: POPC:cholesterol:PEG-PE (7:3:0.15 in μmol) and 100 μg of resveratrol dissolved in ethanol was added to the lipid/chloroform solution. Then, PEG-PE (0.15 μmol), TPP-DSPE-PEG (0.15 μmol) or DQA-DSPE-PEG (0.15 μmol) was added to obtain non-targeting liposomes carrying resveratrol (LS (Res)) and mitochondria-targeting liposomes carrying resveratrol (TLS (Res) or DLS (Res)), respectively. The organic solvents (chloroform and ethanol) were removed by vacuum evaporation, then the dried lipid film was mixed with 1 mL HBG buffer (10 mM HEPES, 5% glucose, pH 7.4) and incubated at room temperature for 4 h with intermittent shaking. The lipid-contained suspension was extruded 11 times through polycarbonate membranes with a 100-nm pore size by a hand-held extruder (Avestin, Ottawa, Canada).

### 2.4. Measurement of Size Distribution and Zeta Potential

To measure the average size and surface charge of LS, TLS and DLS, each sample was diluted 1:100 in HBS (10 mM HEPES, 150 mM NaCl, pH 7.4) to a total volume 1 mL. Then dynamic light scattering (DLS) and electrophoretic light scattering (laser Doppler) analysis of LS, TLS and DLS were performed using a zeta potential and particle size analyzer (ELSZ-1000, Otsuka Electronics Co, Osaka, Japan). The LS (Res), TLS (Res) and DLS (Res) were passed over a Sephadex PD-10 column equilibrated with phosphate buffered saline (137 mM NaCl, 2.7 mM KCl, 8 mM Na_2_HPO_4_ and 2 mM KH_2_PO_4_, PBS, pH 7.4) to remove the non-encapsulated resveratrol.

### 2.5. In Vitro Release Study

In vitro drug release was investigated using previously reported methods with minor modifications [42]. The LS (Res) solution (200 μL of 100 μg/mL resveratrol in liposomes) was prepared and added to a GeBaFlex-tube with a molecular weight cut off of 8 kDa (Gene Bio-Application Ltd., Yavne, Israel). The tubes contained with LS (Res) solution were immersed in 3 mL PBS (pH 7.4) and incubated at 37 °C with rotation at 50 rounds per minute (rpm). Samples of the dissolution medium (3 mL) were collected at various time points (1, 2, 3, 4, 6, 9, 12, 24 and 48 h), and replaced with 3 mL fresh medium at 37 °C. The amount of resveratrol released was assessed using LC/MS/MS analysis, and the samples were prepared by extracting twice with ethyl acetate, followed by drying of the organic phase under high vacuum. The dried residue was then suspended in 20 μL of an acetonitrile and 0.1% formic acid 60:40 (*v*/*v*) mixture and sonicated for 10 min. Finally, a 10 μL aliquot was analyzed using LC/MS/MS, as described below.

### 2.6. In Vitro Cytotoxicity Assay

The cytotoxicity of the LS, TLS, DLS, free Res, LS (Res), TLS (Res) and DLS (Res) was assessed using the cell viability assay. Briefly, B16F10 were seeded in 96-well plates at a density of 1 × 10^4^ cells/well and incubated overnight at optimal condition. The cells were treated by replacing the medium with fresh serum-free medium containing a range of concentrations of LS, TLS, DLS, free Res, LS (Res), TLS (Res) and DLS (Res). Following incubation for 24 h at 37 °C, the cells were washed twice with PBS, then incubated with serum-free medium containing water-soluble tetrazolium (WST) solution (Ez-Cytox Cell Viability Assay Kit, DoGen, Korea) for 30 min at 37 °C in the dark. The absorbance was measured at 480 nm using a microplate reader (Epoch, BioTek Instruments, Winooski, VT, USA).

### 2.7. In Vitro Cellular Uptake Analysis by Laser Scanning Confocal Microscopy

The B16F10 murine melanoma cells were maintained in Dulbecco’s Modified Eagle’s medium (DMEM) supplemented with 10% fetal bovine serum (complete DMEM medium) in a humidified incubator under 5% CO_2_. The cells at a density of 1 × 10^5^ cells were seeded on coverslips (12 mm, Fisher Scientific) and incubated for 24 h at 37 °C and in a 5% CO_2_ atmosphere. The LS, TLS and DLS, containing rhodamine-labeled PE (Rh-PE), were diluted to a final concentration of 100 μg/mL in serum-free medium and treated to the cells. After 1-h incubation, the cells were stained with 4′,6-diamidino-2-phenylindole (DAPI) and fixed using 2% (*w*/*v*) paraformaldehyde in PBS. The coverslips were mounted with a mounting medium (Fluoromount, Sigma-Aldrich) and the fluorescent images of cells on the coverslip were analyzed using a laser scanning confocal microscope (LSCM, A1Plus, Nikon, Tokyo, Japan).

### 2.8. In Vitro Quantification of Mitochondrial Accumulation of Resveratrol by LC–MS/MS

#### 2.8.1. Treatment of Liposomes and Cell Fractionation

B16F10 cells were grown with complete DMEM medium in a 100-mm dish overnight to 80% confluence. Free Res, LS (Res), DLS (Res) and TLS (Res) were diluted with serum free media to total 5 mL, corresponding to 20 µg resveratrol, and treated to the cells. After 10 min, 30 min and 1 h of incubation at 37 °C, the cells were harvested, collected by centrifugation and two times washed with PBS. To separate mitochondria in the cells, cell fractionation was performed on the cell pellet using the Mitochondria/Cytosol Fractionation kit (Biovision, USA). Briefly, 1 mL of 1× cytosol extraction buffer mix containing DTT and protease inhibitors was added to the pellet (according to the manufacturer’s protocol) and mixed and incubated on ice for 10 min. Next, the cells were homogenized on ice using a glass tissue grinder, then the homogenate was centrifuged at 3000 rpm for 10 min at 4 °C. The supernatant was replaced to the new Ep tube and then centrifuged at 13,000 rpm for 30 min at 4 °C. The supernatant in this step contains the cytosol fraction. Then the pellet was resuspended in the mitochondrial extraction buffer mix, containing DTT and protease inhibitors. The mitochondrial and cytosolic fractions were used to measure the resveratrol taken-up by the mitochondria. Ethylacetate extraction was performed to collect the resveratrol.

#### 2.8.2. LC–MS/MS Conditions

The LC–MS/MS system consists of an Agilent LC 1100 series (Agilent Technologies, CA, USA) binary pump, a vacuum degeneration unit and an auto-sampler system connected to a 6490 triple quadrupole MS equipped with an Agilent jet stream technology electrospray ionization (ESI) source. Chromatographic separation was performed by an analysis Sepax BR-C18 (5 µm, 120 Å 1.0 × 100 mm) column. The maintaining temperature of column was set at 30 °C. The temperature of the auto-sampler was set at 4 °C. Of the sample solution 2 µL was injected and the analytes were eluted under the isocratic condition with acetonitrile and 0.1% formic acid in water (60%:40%, *v*/*v*) pumped at a constant flow of 0.10 mL/min. To detection of the resveratrol in the analytes, the MS/MS system was performed under negative ESI and the multiple reactions monitoring (MRM) mode. The MS operational parameters were: Argon as a collision gas; capillary voltage at 5 kV; gas temperature at 225 °C; gas flow 14.1 I/min; nebulizing gas at 40 psi; collision energies of 18 for resveratrol and the precursor to product ion transitions of 184.8–226.9 *m/z* for resveratrol.

### 2.9. In Vitro ROS Production

The B16F10 cells were seeded on the six-well plates at a density of 4 × 10^5^ cells/well. Then the cells were treated with serum-free culture medium (as the control), free Res, LS (Res), TLS (Res) and DLS (Res), respectively. The final concentration of resveratrol in each treatment sample was 4 μg/mL. After 12 h, the cells were collected, suspended and incubated with DCFH-DA (10 μM) at 37 °C for 20 min. After two washes with cold PBS, the DCF fluorescence was determined using a fluorescence-activated cell sorter (FACS).

### 2.10. In Vitro Mitochondrial Depolarization

The JC-1 was used for the measurement of mitochondrial membrane potential (ΔΨm) due to the ability to shift of the cationic lipophilic fluorochrome JC-1 from red to green. The B16F10 cells were seeded on the six-well plates at a density of 4 × 10^5^ cells/well and treated with serum-free culture medium (as control), free Res, LS (Res), TLS (Res) and DLS (Res; 4 μg/mL as final concentration of resveratrol) for 12 h. The cells were harvested, suspended and incubated with the JC-1 dye (1 μg/mL) in the dark at 37 °C for 10 min. The cells were washed with PBS and fluorescence intensity from cells was measured by flow cytometry.

### 2.11. Statistical Analysis

All of the studies were performed in triplicate. The results were expressed as mean ± standard error of the mean (S.E.M.). The statistical significance of the data was analyzed by Student’s *t*-test and analysis of variance (ANOVA) with Bonferroni’s post-hoc test. A value of *p* < 0.05 was considered to be statistically significant.

## 3. Results and Discussion

### 3.1. Synthesis of TPP-DSPE-PEG and DQA-DSPE-PEG Conjugate

For the targeted delivery of resveratrol to the mitochondria in cancer cells, we proposed liposomes, surface-modified with TPP or DQA. To achieve this, 4-carboxybutyl triphenylphosphonium bromide- polyethylene glycol-distearoylphosphatidylethanolamine (TPP-DSPE-PEG) and dequalinium-polyethylene glycol-distearoylphosphatidylethanolamine (DQA-DSPE-PEG) were synthesized as mitochondriotropic molecules. Their synthesis scheme is presented in Figure 2.

TPP-DSPE-PEG formation was confirmed by the proton NMR spectrum. The appearance of signals at 7.46–7.76 ppm was attributed to the presence of CH=CH and the phenyl ring of TPP and signals at 2.66, 2.0 and 1.6 ppm were attributable to the methylene group of the carboxy butyl linker. Signals at 6 and 6.5 ppm appeared due to the presence of the NH group in the conjugate. In addition, the major signals of DSPE-PEG (at 3.7 ppm for –CH_2_–CH_2_–O of PEG and at 1–3.5 ppm and 0.85 ppm for methylene and the methyl groups in the lipid chain, respectively) in the proton NMR spectrum also indicated the formation of the conjugate (Figure 3A,B). The MALDI-TOF spectrum showed a peak at 3133.5 *m*/*z* that was assigned to TPP-DSPE-PEG (M-[Br^−^] + [2H^+^]; Figure 3C). The MALDI-TOF assay results also clearly confirmed the graft reaction of TPP to DSPE-PEG.

The formation of DQA-DSPE-PEG was confirmed by the proton NMR spectrum. The appearance of signals at 7.7–9.0 ppm and 4.4 ppm were attributed to the presence of CH=CH in the aromatic ring and amine group of DQA, respectively, and signals at 1.2–3.5 ppm were attributable to the methylene group of the aliphatic chain between the two quinolines. Upon formation of the conjugate and the consequent creation of a NH group, a new signal at 6 ppm appeared. In addition, the major signals of DSPE-PEG (at 3.5 ppm for –CH_2_–CH_2_–O of PEG, at 1–3.7 ppm and 0.8 ppm for methylene and the methyl groups in the lipid chain, respectively) in the proton NMR spectrum also indicated the formation of the conjugate (Figure 4A,B). MALDI-TOF spectrum showed a peak at 3342.6 m/z that was assigned to DQA-DSPE-PEG (M-[NH_2_]; Figure 4C). The MALDI-TOF assay results also clearly confirmed the graft reaction of DQA to DSPE-PEG.

### 3.2. Characterization of TLS (Res) and DLS (Res)

Liposome-based drug delivery systems have been proposed for the delivery of hydrophobic as well as hydrophilic therapeutic molecules and showed high biocompatibility with low toxicity [38]. Considering these benefits, a liposome carrier system was developed in this study to protect and stabilize resveratrol. In addition, the constructed resveratrol-containing liposomes exposed the molecules TPP or DQA on their surface to target specifically the mitochondria. In this study, the mitochondriotropic liposomes were constructed with rational lipid composition and thin-film hydration followed by the extrusion method according to previously published method [33,43], the amount of mitochondrial targeting ligand was decided as the amount to exert a targeting effect [2,44,45]. Table 1 shows the average particle sizes, polydispersity index (PDI) and zeta potentials of LS (Res), TLS (Res) and DLS (Res). Results indicated that the mean hydrodynamic diameter of the liposomes was approximately 120 nm with a narrow size distribution (PDI < 0.3). The narrow and uniform size distributions of LS (Res), TLS (Res) and DLS (Res) were obtained after membrane extrusion. The determined surface charges are −1.68 ± 0.31 mV, 10.46 ± 0.43 mV and 13.79 ± 0.31 mV for LS (Res), TLS (Res) and DLS (Res), respectively (Table 1). TLS (Res) and DLS (Res) showed a positive surface charge, whereas LS (Res) displayed a neutral surface charge, confirming the presence of the highly positively charged TPP or DQA moieties on the liposomes, and demonstrating that they contributed to the change of the liposome’s surface charge. It was reported that the loading capacity of extruded liposomes carrying resveratrol showed more than 5% and the encapsulation efficiencies of resveratrol in liposomal carriers were found to be more than 90% [2,46,47,48]. Based on the references, we prepared the liposomes with a lipids/resveratrol ratio of 20:1 (*w*/*w*). The diameter of approximately 100 nm of TLS (Res) and DLS (Res) should allow accumulation into tumor tissue by the enhanced permeability and retention (EPR) effect [35].

To evaluate whether resveratrol was released by the liposomes, the drug release rate was measured at 37 °C in PBS (pH 7.4). Figure 5 shows the cumulative release profile of resveratrol in percentage. The resveratrol-loaded liposomes showed an initial relatively faster release (31.11% ± 4.15%) in PBS (pH 7.4) within 4 h, followed by a sustained release pattern (62.12% ± 10.25%) up to 48 h, whereas free resveratrol showed a release burst with ca. a 100% release at 2 h. The liposomal system demonstrated a sustained release profile, useful for the sustained delivery of resveratrol.

### 3.3. In Vitro Cellular Cytotoxicity

Figure 6A shows the cytotoxicity of the non-targeting (LS) and mitochondria-targeting liposomes (TLS and DLS) without resveratrol in B16F10 cells. Cell viability showed no significant differences among cells treated with LS, TLS and DLS for 24 h. The average cell viability showed a higher than 90% up to the liposomes concentration of 500 μg/mL. This result indicated that mitochondriotropic liposomes did not seriously damage cells, demonstrating that the liposomes itself did not contribute to the cellular toxicity.

To further evaluate the cytotoxicity of the mitochondria-targeting liposomes carrying resveratrol, an MTT assay was carried out with different concentrations of the liposomes. As shown in Figure 6B, the free Res, LS (Res), TLS (Res) and DLS (Res) showed dose-dependent cytotoxicity against B16F10 cells. In particular, the mitochondriotropic liposomes carrying resveratrol (TLS (Res) and DLS (Res)) showed the lower viability than free Res and LS (Res), indicating that the mitochondria-targeting liposomes improves the cytotoxicity efficacy of resveratrol in cancer cells.

### 3.4. In Vitro Cellular Uptake and Mitochondrial Targeting

Figure 7 represents the laser scanning confocal microscopic images of the cellular uptake and intracellular localization of liposomes in B16F10 cells. To observe this, the non-targeting (LS) and mitochondria-targeting liposomes (TLS and DLS) without resveratrol were labeled with rhodamine-labeled PE (red fluorescence) and the mitochondria were stained by MitoTracker green (green fluorescence). The red fluorescence intensity of TLS and DLS in the B16F10 cells after 1 h of incubation was stronger than that of LS, indicating higher cellular uptake of TLS and DLS into B16F10 cells compared to LS. The positive surface charge of TLS and DLS may contribute to an easier attachment to and uptake into the cells, as they have a negative surface charge due to the phospholipid bilayer composition [49]. Additionally, a bright yellow fluorescence, which is the result of the overlay of red and green fluorescence, was observed for TLS and DLS, indicating that TLS and DLS were localized on the mitochondria after internalization by the cells. In the case of LS, however, their mitochondrial localization was reduced compared to TLS and DLS. These results demonstrate that mitochondria-targeting liposomes labeled with TPP or DQA are able to accumulate selectively in the mitochondria.

Furthermore, to quantify the mitochondrial accumulation of resveratrol delivered by mitochondria-targeting liposomes, mitochondrial and cytosol fractionation was performed. The amount of mitochondrial accumulation was quantified using LC–MS/MS and the percentage ratio was calculated based on total uptake amount of resveratrol in each treated group. At 1 h post-incubation, significantly higher concentrations of resveratrol were found in the cells treated with TLS (Res; 785.3 ± 82.7 ng/mL) and DLS (Res; 927.1 ± 101.2 ng/mL) as compared to those treated with LS (Res; 418.0 ± 31.9 ng/mL) and free resveratrol (227.5 ± 58.3 ng/mL; data not shown). As shown in Figure 8, the mitochondrial resveratrol percentage ratio of TLS (Res) and DLS (Res) was 2.4- and 3.1-fold higher than that of LS (Res) in B16F10. In addition, the mitochondrial resveratrol percentage ratio of TLS (Res) and DLS (Res) was 5.1- and 6.6-fold higher than that of free resveratrol in B16F10. These results indicate that TLS (Res) and DLS (Res) were selectively accumulated in the mitochondria by the TPP- and DQA-moieties on the surface of liposomes after internalization by the cells.

### 3.5. In Vitro ROS Production

To determine the effect of resveratrol delivered by mitochondrial-targeting liposomes, we evaluated the ROS generation in the B16F10 cells treated with the free resveratrol, LS (Res), TLS (Res) and DLS (Res) after 12 h of incubation in the presence of the DCFH-DA reagent using FACS. The DCFH-DA fluorescent dye was used as it is transformed into 2,7-dichlorofluorescein upon ROS generation. The positive control was used as a standard of occurring change in the ROS production. As shown in Figure 9, fluorescence intensity of LS (Res), TLS (Res) and DLS (Res) increased more than that of free resveratrol in B16F10 cells after 12 h incubation, indicating that liposome formulations generated greater intracellular ROS levels compared to free resveratrol. The mean of fluorescence intensity in each group was calculated as TLS (Res; 1024.5 ± 12.3), DLS (Res; 1799.1 ± 14.9), LS (Res; 842.2 ± 7.9), free resveratrol (705.3 ± 17.1) and positive control (619.3 ± 25.1). TLS (Res) and DLS (Res) generated 1.2- and 2.1-fold greater ROS levels than LS (Res) did. In contrast, no ROS was generated in the control cells without drug treatment. ROS production in the mitochondria is closely associated with cell damage, such as inflammation, oxidative stress and cell organelle disruption [50,51,52].

### 3.6. In Vitro Mitochondrial Depolarization

To evaluate the mitochondrial depolarization, flow cytometry studies with JC-1 dye staining were performed in B16F10 cells treated with free resveratrol, LS (Res), TLS (Res) and DLS (Res) for 12 h. A switch from the red to green fluorescence of the JC-1 dye indicated depolarization. The negative control and positive control indicate cells stained without and with JC-1, respectively. The two control cells were incubated in only serum free media for 12 h. The positive control presents only JC-1-stained the cells without any treatments. The degree of mitochondria depolarization was evaluated by moved events from the upper right to lower right section of the diagrams. As shown in Figure 10, the percentage of mitochondrial membrane depolarization was 7.95% ± 1.32%, 13.41% ± 2.95%, 24.78% ± 4.19% and 32.45% ± 3.07% in the free resveratrol, LS (Res), TLS (Res) and DLS (Res)-treated cells, respectively. The mitochondrial membrane potential (ΔΨ_m_) of the TLS (Res) and DLS (Res)-treated cells was 1.8- and 2.4- fold higher than that of the LS (Res)-treated cells. The TLS (Res) and DLS (Res)-treated cells had the biggest dissipation of ∆Ψ_m,_ indicating that the mitochondria-specific targeting of resveratrol, by TPP and DQA moieties on the liposomes’ surface, triggered the increased dissipation of ∆Ψ_m_. It was reported that ROS production can be triggered by mitochondria dysfunction [53] and excess levels of ROS in the cells cause damage to cellular organelles, which can lead to activation of apoptosis [54,55]. The mitochondrial membrane depolarization is the result of mitochondrial dysfunction, which leads to cell death via initiation of the apoptotic pathway.

## 4. Conclusions

In this study, TPP-DSPE-PEG and DQA-DSPE-PEG were synthesized as mitochondriotropic molecules with the aim of enhancing both the mitochondrial targeting of nanoparticle systems and the anticancer efficacy of the resveratrol. The conjugates were characterized using physical methods including NMR and MS. The mitochondria-targeting liposomes carrying resveratrol were constructed by modifying the TPP-DSPE-PEG or DQA-DSPE-PEG on the surface of liposomes. TLS (Res) and DLS (Res) showed superior in vitro behavior in the B16F10 cells, including increased accumulation in mitochondria, anticancer efficacy, ROS generation and mitochondrial depolarization, compared to that of LS (Res). Since the selectivity of liposomes for cancer cells can be improved by adding a cancer-targeting moiety on their surface, the mitochondria-targeting liposomes carrying resveratrol provide a potential strategy for cancer treatment by mitochondrial targeting delivery of therapeutics and stimulation the mitochondrial signaling pathway.

## Figures and Tables

**Figure 1 pharmaceutics-11-00423-f001:**
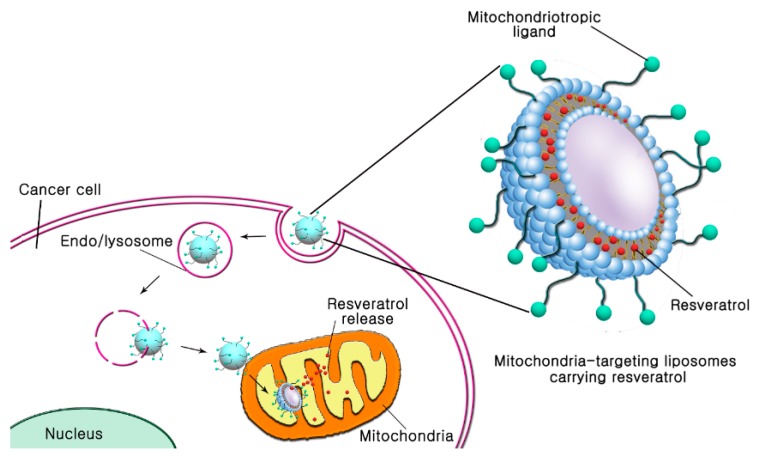
Schematic diagram to describe mitochondriotropic liposomes for enhanced subcellular trafficking of resveratrol in cancer cells. Mitochondria-targeting liposomes carrying resveratrol were constructed by modifying the TPP-DSPE-PEG (TLS (Res)) or DQA-DSPE-PEG (DLS (Res)) on the surface of liposomes.

**Figure 2 pharmaceutics-11-00423-f002:**
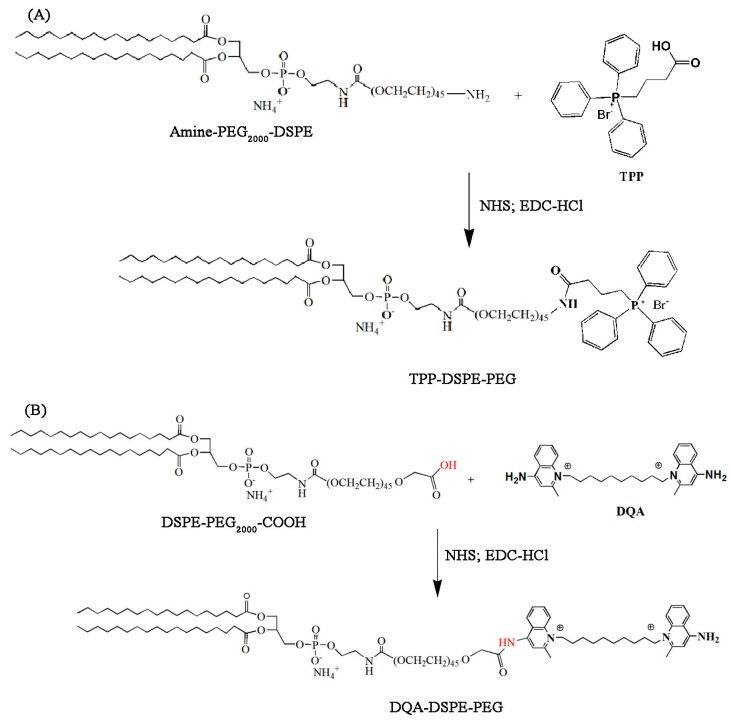
Synthesis scheme of TPP-DSPE-PEG (**A**) and DQA-DPPE-PEG (**B**) conjugate. DSPE-PEG_2000_-COOH, 1,2-distearoyl-*sn*-glycero-3-phosphoethanol-amine-*N*-[methoxy(polyethylene glycol)-2000-COOH] and DQA-DPPE-PEG, dequalinium-polyethylene glycol-distearoylphosphatidylethanolamine; TPP-DSPE-PEG, 4-carboxybutyl triphenylphosphonium bromide- polyethylene glycol-distearoylphosphatidylethanolamine.

**Figure 3 pharmaceutics-11-00423-f003:**
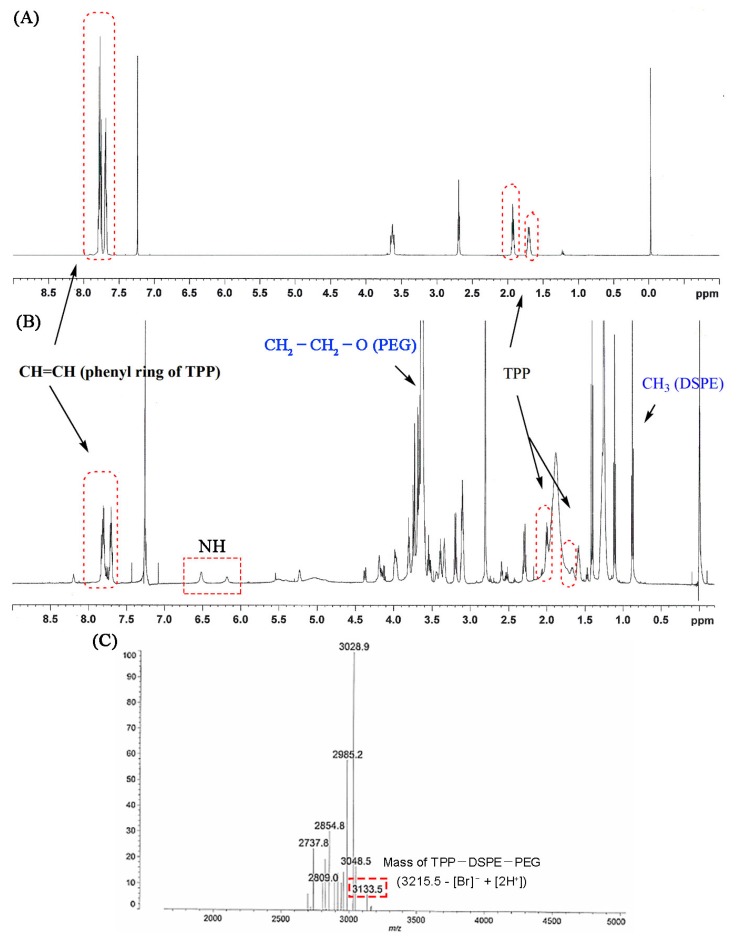
Proton NMR spectra of carboxybutyl triphenylphosphonium bromide (TPP) (**A**) and TPP-DSPE-PEG conjugate (**B**). (**C**) MALDI-TOF mass of TPP-DSPE-PEG.

**Figure 4 pharmaceutics-11-00423-f004:**
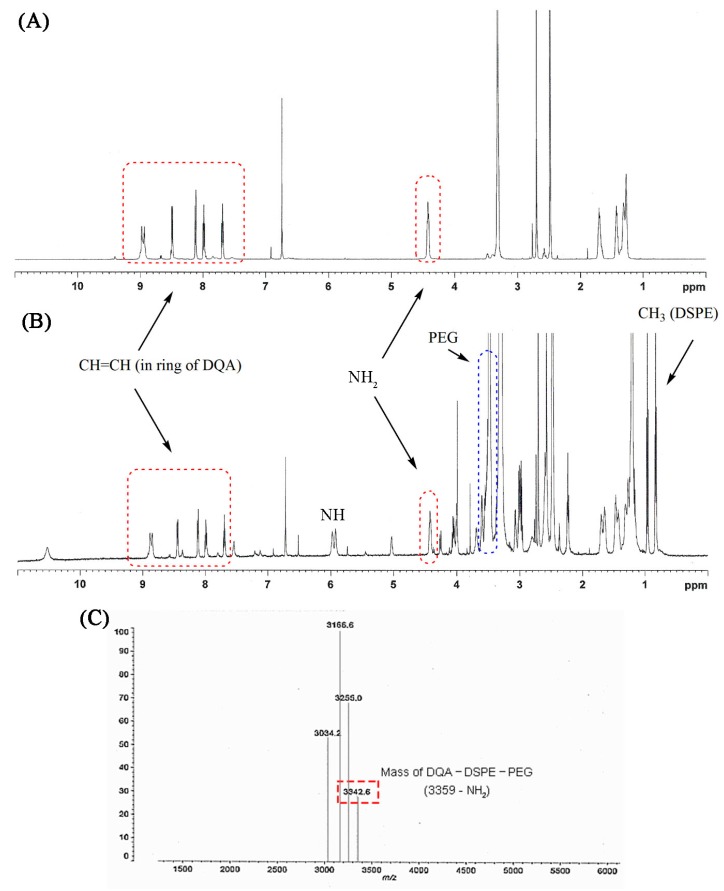
Proton NMR spectra of dequalinium (DQA) (**A**) and DQA-DSPE-PEG conjugate (**B**). (**C**) MALDI-TOF mass of DQA-DSPE-PEG.

**Figure 5 pharmaceutics-11-00423-f005:**
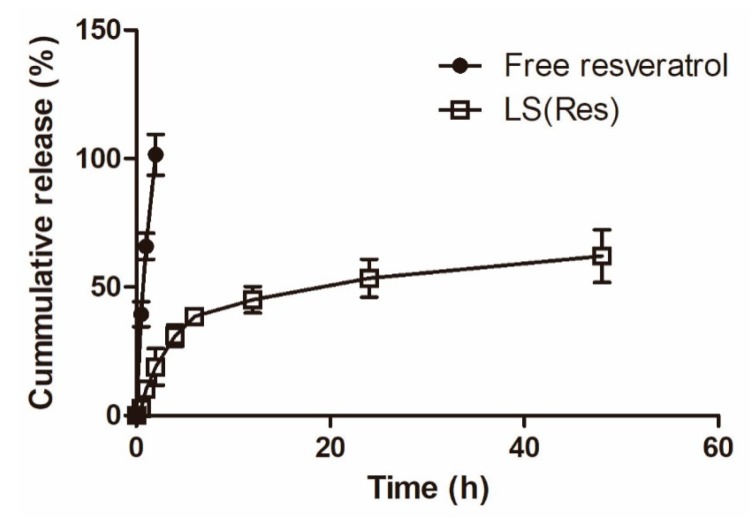
Percentage of cumulative release of free resveratrol and LS (Res) in PBS (pH 7.4). Mean ± S.E.M, *n* = 3.

**Figure 6 pharmaceutics-11-00423-f006:**
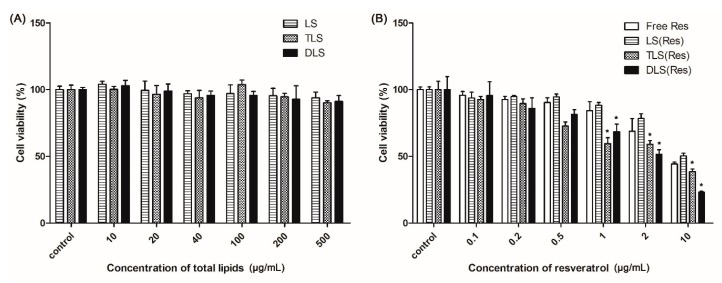
In vitro viability of B16F10 cells after applying the liposomes (**A**) non-carrying resveratrol and (**B**) carrying resveratrol at 24 h. Mean ± S.E.M, *n* = 4. * indicates a statistical difference from the LS (Res), *p* < 0.05.

**Figure 7 pharmaceutics-11-00423-f007:**
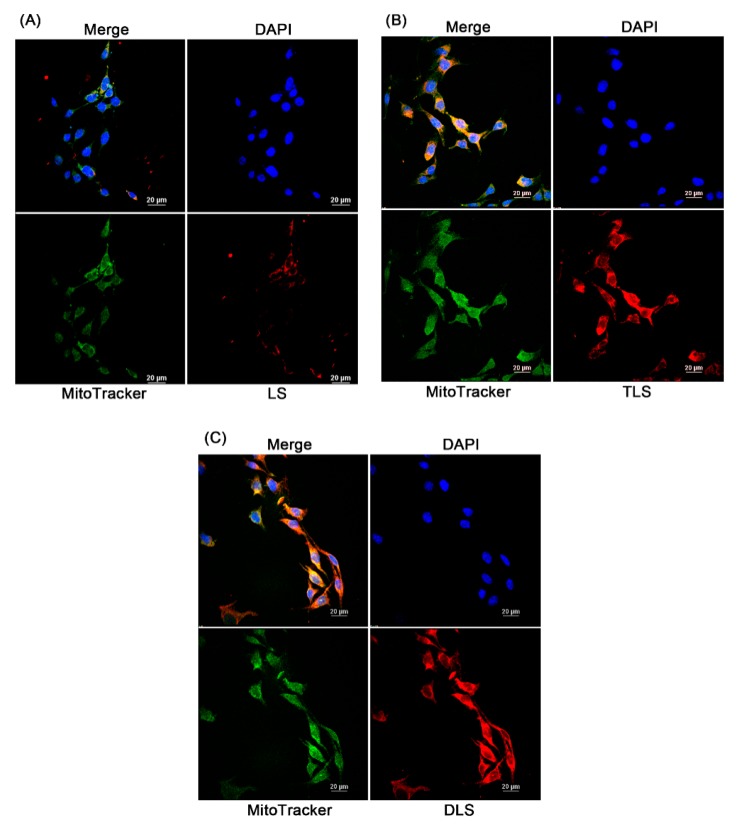
Cellular uptake of the non-targeting liposomes without resveratrol (LS) and mitochondrial-targeting liposomes without resveratrol (TLS and DLS) in B16F10 cells. Laser scanning confocal microscopy (LSCM) images after a 1 h of incubation with LS (**A**), TLS (**B**) and DLS, and (**C**) red, rhodamine-labeled liposome; green, MitoTracker; blue, 4′,6-diamidino-2-phenylindole (DAPI).

**Figure 8 pharmaceutics-11-00423-f008:**
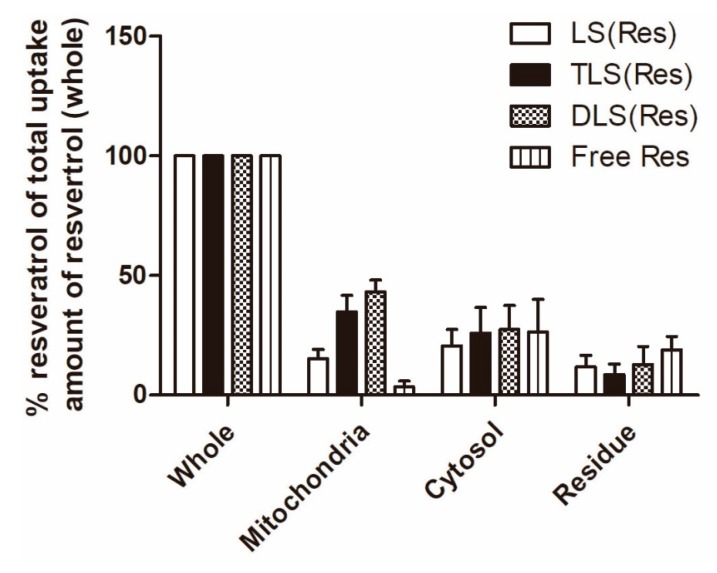
Intracellular uptake of LS (Res), TLS (Res), DLS (Res) and free resveratrol by B16F10 cells after 1 h of incubation. The data represent the mean ± S.E.M (*n* = 3).

**Figure 9 pharmaceutics-11-00423-f009:**
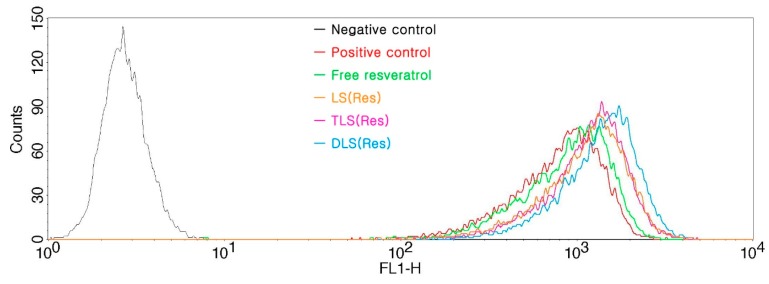
Reactive oxygen species (ROS) production measured with DCFH-DA in B16F10 cells treated with free resveratrol, LS (Res), TLS (Res) and DLS (Res) after 12 h of incubation. Production of ROS was measured in fluorescence at excitation and emission wavelengths of 485 and 528 nm, respectively.

**Figure 10 pharmaceutics-11-00423-f010:**
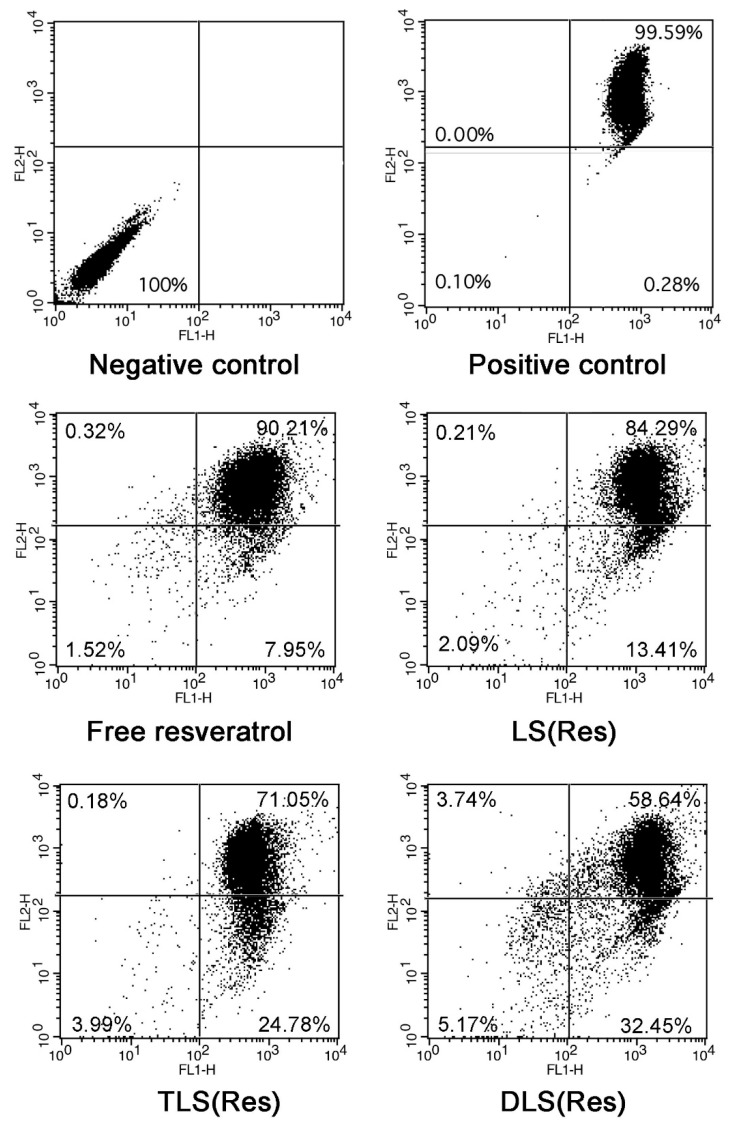
Mitochondrial depolarization of free resveratrol, LS (Res), TLS (Res) and DLS (Res) after 12 h incubation using flow cytometry.

**Table 1 pharmaceutics-11-00423-t001:** Size, polydispersity index (PDI) and zeta potential of LS (Res), TLS (Res) and DLS (Res). Mean ± S.E.M, *n* = 5.

Formulations	Size Distribution (nm)	PDI	Zeta Potential (mV)
LS (Res)	128.10 ± 6.77	0.25 ± 0.12	−1.68 ± 0.31
TLS (Res)	115.50 ± 3.99	0.22 ± 0.02	10.46 ± 0.43
DLS (Res)	121.18 ± 3.65	0.28 ± 0.02	13.79 ± 0.31

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
