# Peer review of "Enhanced Subcellular Trafficking of Resveratrol Using Mitochondriotropic Liposomes in Cancer Cells"

_pharmaceutics, 2019, doi:10.3390/pharmaceutics11080423_

Round 1

Reviewer 1 Report

The manuscript reports a study on the effect of resveratrol (Res) loaded mitochondriotropic liposomes on B16F10 murine melanoma cells and concludes that these liposomes "could provide a potential strategy to treat cancers by inducing apoptosis via the mitochondrial signaling pathway" (abstract, line 24).

The basic drawbacks of this study is that a) it does not report any data on cell viability (toxicity) of the proposed formulations, and b) it does not examine if apoptosis (as claimed in the abstract) or necrosis is induced by them employing the specific protocols for this. Moreover, given that, in the literature, liposomes and in general nanoparticles decorated with mitochondriotropic agents are often reported to show toxicity per se, significant control experiments are missing, namely the effect of empty TLS and DLS in all reported experiments. Finally, there is practically no discussion section. The authors should discuss their findings as accustomed.

In addition to the above, the following remarks should be considered:

In Fig. 1 and throughout the manuscript the authors indicate that liposomes remain intact after entering the cell through the endosomal/lysosomal pathway and, still intact, reach and penetrate the mitochondrion. According to the literature this is highly unlikely to occur and certainly there are no data to support this. The data presented show that either rhodamine-PE and Res can be found inside the mitochondrion - not the intact liposome.

To justify the choice to use Res for chemoprevention in this study, the authors cite only two references that are merely dealing with an vitro cell and an in vivo animal study (2,27). This part should be enriched with more papers, and preferably with studies of Res in clinical trials.

The synthesis of TPP-PEG and DQA-PEG conjugates (part 2.2.) must be corrected. In the first case they use DSPE-PEG-NH2 and in the second DSPE-PEG-COOH. As it is written, it seems as if they use in both cases DSPE-PEG-COOH. Also, there is no purification step reported after the reactions (only filtering through a 0.45 um filter). This implies that unreacted DSPE-PEG-NH2 or DSPE-PEG-COOH are also present.

For the preparation of Res-loaded liposomes there is no mention of removing the non-encapsulated Res. In the literature the solubility of Res in water is ~30 ug/mL so a significant amount of Res (total Res added is 100 ug) must be non-encapsulated in the aqueous phase and this should have been removed.

What was the reason to use POPC, and not for instance DPPC, for the preparation of liposomes? In the materials section they mention also POPG but they do not refer to it again in the manuscript. Did they use it as well for the preparation of liposomes or not?

In the in vitro release studies did they observed any other molecules (such as DSPE-PEG-COOH or –NH2) being released?

The proton NMR spectra must be fully assigned and 13C NMR spectra must be included. Emphasis should be given to the formation of the new amide bond. It is not clear why 2 different NH2 peaks are observed. This could imply that non-reacted DSPE-PEG-NH2 is present. Also the 13C NMR spectra are needed to provide proof that no unreacted DSPE-PEG-COOH is present and that the amide bond is actually formed in both cases.

In the confocal microscopy section the authors should made it clear in the text that they observe Rhodamine-PE and not liposomes. Rhodamine by itself is targeting mitochondria so it is normal to observe its fluorescence located there in all cases. The photographs presented lack detail and clarity. Even the MitoGreen is shown all over the cytosol and not in distinct mitochondria. Improved photos are needed to clearly show the mitochondria in the cytosol before concluding that there is co-localization of green and red –which is normal since both compounds are known to be located there.

The Res accumulation in mitochondria, cytosol and cells is expressed as ng/mL. What does mL stands for? Normally in this kind of experiments the results are expressed with respect to protein content in each sample (in each dish or well) and not to the volume of the aqueous phase. Also it would be interesting to know if TPP-PEG or DQA-PEG was also detected in this experiment in mitochondria or cytosol.

For the experiments employing FACS (after 12 h incubation time) did the authors collected only the living cells or all cells?

The curves shown in Fig. 8 are quite difficult to follow, especially to distinguish the reddish lines. In any case, it seems as if even the simple Res loaded liposomes are very similar to the positive control or the free Res.

There is no explanation what does positive control stands both for ROS production and mito depolarization experiments.

Especially for the mito depolarization experiments it seems that LS(Res) and free Res is very similar to the positive control (loss of mitochondrial depolarization) as all events are shown in the upper right section of the diagrams.

Author Response

Response to Reviewer 1 Comments

The manuscript reports a study on the effect of resveratrol (Res) loaded mitochondriotropic liposomes on B16F10 murine melanoma cells and concludes that these liposomes "could provide a potential strategy to treat cancers by inducing apoptosis via the mitochondrial signaling pathway" (abstract, line 24).

The basic drawbacks of this study is that a) it does not report any data on cell viability (toxicity) of the proposed formulations, and b) it does not examine if apoptosis (as claimed in the abstract) or necrosis is induced by them employing the specific protocols for this. Moreover, given that, in the literature, liposomes and in general nanoparticles decorated with mitochondriotropic agents are often reported to show toxicity per se, significant control experiments are missing, namely the effect of empty TLS and DLS in all reported experiments.

Author’s response:

In this study, we mainly focused on comparison of mitochondria targeting ability with two mitochondriotropic liposomes. To verify the mitochondria targeting ability of the liposomes, we carried out in vitro experiments focused on mitochondria accumulation of drug, mitochondria stimulation and mitochondria dysfunction, not toxicity. To clarify our main objective, the abstract was modified as followed:

“Our results demonstrated that TLS(Res) and DLS(Res) could provide a potential strategy to treat cancers by mitochondrial targeting delivery of therapeutics and stimulation of the mitochondrial signaling pathway.”

Finally, there is practically no discussion section. The authors should discuss their findings as accustomed.

Author’s response:

As per the comment, in-depth discussions on our findings were provided in results and discussion section of the revised manuscript.

In addition to the above, the following remarks should be considered:

In Fig. 1 and throughout the manuscript the authors indicate that liposomes remain intact after entering the cell through the endosomal/lysosomal pathway and, still intact, reach and penetrate the mitochondrion. According to the literature this is highly unlikely to occur and certainly there are no data to support this. The data presented show that either rhodamine-PE and Res can be found inside the mitochondrion - not the intact liposome.

Author’s response:

As pointed out, it is not clear whether the liposomes remain intact or not after entering the cell.  However, studies related with successful liposomal delivery for the cellular trafficking through eodosomal escape have been reported in many research of drug delivery systems. Our liposomal system was also well established for the endosomal escape and researches related with that were already reported (JCR 2009, Ko, et al., Macromol. Rapid Commun 2010 Ko, et al., AAPS PharmSciTech 2014, Kang, et al.).

To justify the choice to use Res for chemoprevention in this study, the authors cite only two references that are merely dealing with an vitro cell and an in vivo animal study (2,27). This part should be enriched with more papers, and preferably with studies of Res in clinical trials.

Author’s response:

As suggested, the part was enriched with more papers related with studies of resveratrol in clinical trials (reference 28-30).

The synthesis of TPP-PEG and DQA-PEG conjugates (part 2.2.) must be corrected. In the first case they use DSPE-PEG-NH2 and in the second DSPE-PEG-COOH. As it is written, it seems as if they use in both cases DSPE-PEG-COOH. Also, there is no purification step reported after the reactions (only filtering through a 0.45 um filter). This implies that unreacted DSPE-PEG-NH2 or DSPE-PEG-COOH are also present.

Author’s response:

As per the comments, the synthesis of TPP-PEG and DQA-PEG conjugates (part 2.2.) was corrected and the purification step was added as followed:

“DSPE-PEG2000-COOH (25 mg, 0.0088 mmol) or DSPE-PEG2000-NH2 (25 mg, 0.0088 mmol) was allowed to react for 3 h with EDC-HCl (3.4 mg, 0.018 mmol) and NHS (1.01 mg, 0.0087 mmol) in 2 mL of chloroform in the presence of 3 drops of triethylamine at room temperature in the dark. The progress of the reaction was checked by thin-layer chromatography (TLC) on a silica gel plate. Next, dequalinium (DQA) (4.6 mg, 0.0087 mmol) or TPP (3.9 mg, 0.0087 mmol) dissolved in DMSO (2 mL) were added to DSPE-PEG2000-COO-NHS and stirred for 30 min. Chloroform was removed by an evaporator, and 5 mL of distilled water was added. The final mixture was dialyzed using a cellulose dialysis tubing (MWCO 3000) against deionized water for 48 h, and then freeze-dried using a lyophilizer. After freeze-drying, the residue was redissolved in distilled water and filtered through a 0.45-µm filter syringe. The liquid was then freeze-dried again and the formed product was characterized by proton NMR spectroscopy (Brucker, 600 MHz, Billerica, MA) and MALDI-TOF mass spectrometer (AXIMA-Assurance, Shimadzu, Kyoto, Japan).”

For the preparation of Res-loaded liposomes there is no mention of removing the non-encapsulated Res. In the literature the solubility of Res in water is ~30 ug/mL so a significant amount of Res (total Res added is 100 ug) must be non-encapsulated in the aqueous phase and this should have been removed.

Author’s response:

As pointed out, the procedure for removing non-encapsultaed resveratrol was included as followed:

“The LS(Res), TLS(Res) and DLS(Res) were passed over a Sephadex PD-10 column equilibrated with phosphate buffered saline (137 mM NaCl, 2.7 mM KCl, 8 mM Na2HPO4, and 2 mM KH2PO4, PBS, pH 7.4) to remove the non-encapsulated resveratrol.”

What was the reason to use POPC, and not for instance DPPC, for the preparation of liposomes? In the materials section they mention also POPG but they do not refer to it again in the manuscript. Did they use it as well for the preparation of liposomes or not?

Author’s response:

POPC is among the primary constituents of cellular membranes. The POPC is the most abundant and has been extensively used as a model compound for representation of natural PC mixtures. Therefore, POPC was used to develop nanoparticle system with higher biologic compatibility for pharmaceutical applications (Nanoscale research letters, 2013, Akbarzadeh et al., Drug Carriers—Advances in Research and Application: 2013 Edition, Acton) "

As pointed out, POPG was not used in the process of preparation of liposomes. Therefore, the POPG mentioned in the materials section was deleted.

In the in vitro release studies did they observed any other molecules (such as DSPE-PEG-COOH or –NH2) being released?

Author’s response:

Although the in vitro releases of surface-modified liposomes were not tested, it can be expected that the targeting ligands on the surface would not exert any significant effects on the release of the cargo, thus providing comparable release profiles to those of unmodified liposome.

 The proton NMR spectra must be fully assigned and 13C NMR spectra must be included. Emphasis should be given to the formation of the new amide bond. It is not clear why 2 different NH2 peaks are observed. This could imply that non-reacted DSPE-PEG-NH2 is present. Also the 13C NMR spectra are needed to provide proof that no unreacted DSPE-PEG-COOH is present and that the amide bond is actually formed in both cases.

Author’s response:

Although 13C NMR could tell whether any unreacted DSPE-PEG-COOH or DSPE-PEG-NH2 are remaining, it should be noted that the reaction mixtures were purified to remove the unreacted molecules and thus proton NMR could provide sufficient evidences to confirm the synthesis of the conjugates by the formation of the new amide bond.  

In the confocal microscopy section the authors should made it clear in the text that they observe Rhodamine-PE and not liposomes. Rhodamine by itself is targeting mitochondria so it is normal to observe its fluorescence located there in all cases. The photographs presented lack detail and clarity. Even the MitoGreen is shown all over the cytosol and not in distinct mitochondria. Improved photos are needed to clearly show the mitochondria in the cytosol before concluding that there is co-localization of green and red –which is normal since both compounds are known to be located there.

Author’s response:

Although rhodamine by itself is able to target mitochondria, the effect of rhodamine was minimal in our experimental condition as shown by the lower mitochondrial accumulation of LS than those of TLS or DLS.

Although the MitoGreen staining was not distinctive, it was sufficient to provide qualitative comparison between LS and TLS or DLS. It should also be noted that quantitative evaluation on mitochondrial accumulation was provided by LC-MS/MS analysis in Figure 7.

The Res accumulation in mitochondria, cytosol and cells is expressed as ng/mL. What does mL stands for? Normally in this kind of experiments the results are expressed with respect to protein content in each sample (in each dish or well) and not to the volume of the aqueous phase. Also it would be interesting to know if TPP-PEG or DQA-PEG was also detected in this experiment in mitochondria or cytosol.

Author’s response:

The resveratrol was extracted by liquid-liquid-extraction (LLE) from the mitochondria, cytosol and cells fraction suspended in buffer and quantified using LC-MS/MS, thus resveratrol accumulation in mitochondria, cytosol and cells was expressed as concentration unit.

The TPP-PEG or DQA-PEG in mitochondria or cytosol were not quantified due to limitation of m/z scan range in LC-MS/MS.

For the experiments employing FACS (after 12 h incubation time) did the authors collected only the living cells or all cells?

Author’s response:

It should be noted that only the living cells were collected in the FACS analysis.

The curves shown in Fig. 8 are quite difficult to follow, especially to distinguish the reddish lines. In any case, it seems as if even the simple Res loaded liposomes are very similar to the positive control or the free Res.

Author’s response:

Although there is the difference between the LS(Res) and positive control or free Res, it is difficult to follow that due to the log scale of fluorescence intensity in the x-axis, thus quantified results of fluorescence intensity in each group were described in the results section as followed:

“The mean of fluorescence intensity in each group was calculated as TLS(Res) (1024.5 ± 12.3), DLS(Res) (1799.1 ± 14.9), LS(Res) (842.2 ± 7.9), free resveratrol (705.3 ± 17.1) and positive control (619.3 ± 25.1).”

There is no explanation what does positive control stands both for ROS production and mito depolarization experiments.

Author’s response:

As per the comment, the sentences related with positive control were added in the result section as followed:

“The positive control was used as standard of occurring change in the ROS production.”

“The negative control and positive control indicate cells stained without and with JC-1, respectively. The two control cells were incubated in only serum free media for 12 h. The positive control presents only JC-1-stained the cells without any treatments.”

Especially for the mito depolarization experiments it seems that LS(Res) and free Res is very similar to the positive control (loss of mitochondrial depolarization) as all events are shown in the upper right section of the diagrams.

Author’s response:

The degree of mitochondria depolarization was evaluated by moved events from the upper right to lower right section of the diagrams. The movement was observed in all treated group, this indicates that change of mitochondrial membrane potential occurred in all treated groups compared to positive control.

Reviewer 2 Report

Manuscript ID: pharmaceutics-515761

Title: “Enhanced subcellular trafficking of resveratrol using mitochondriotropic liposomes in cancer cells”

The manuscript “Enhanced subcellular trafficking of resveratrol using mitochondriotropic liposomes in cancer cells” is a research paper that address the use a lipid-based nanocarriers to target resveratrol for mitochondria in order to induce apoptosis in cancer cells. The lipid-based nanocarriers are liposomes containing one mitochondriotropic cationic lipid-derivative, namely: Carboxybutyl triphenylphosphonium bromide-polyethylene glycol-distearoylphosphatidylethanolamine (TPP-DSPE-PEG) or dequalinium-polyethylene glycol distearoylphosphatidylethanolamine (DQA-DSPE-PEG).

1.       The mitochondriotropic lipid-derivatives were synthetized considering traditional procedures, and authors use NMR and MALDI-TOF mass to prove that the synthesis of TPP-DSPE-PEG and DQA-DSPE-PEG conjugates was well-succeeded, as shown in figure 3. However, the procedures for NMR and MALDI-TOF mass were not included in the Materials and Methods section. This procedures are important and should be included in the new version of manuscript.

2.     The subsection “2.3. Preparation of TPP-DSPE-PEG-modified liposomes carrying resveratrol (TLS(Res)) and DQA-DSPE-PEG-modified liposomes carrying resveratrol (DLS(Res)” must be rewritten in order to remove the errors related with the concentration units of each component [e.g. The following lipids (total amount of 2 mg) were dissolved in chloroform: POPC (final concentration = 7 μmol) … ]. The amount the components is confused with concentration. The liposomes composition should be indicated considering the molar proportions among all components (e.g. - POPC:Cholesterol: PEG-PE: resveratrol; 7/3/0.15/0.438, in μmoles), which allow to calculate the molar ratio of each one. The final volume of water solution buffer used to prepare the liposomes should be indicated.

3.     The mitochondriotropic liposomes were characterized, considering the size and surface charge, assessed in terms of zeta potential. However, several other properties (and aspects) must be considered to show their potential as a therapeutic tool. Thus, a rationalization of the use of above-mentioned molar proportions of each lipid component (and cationic mitochondriotropic lipid derivative) will be also included in the new version of the manuscript; and additional assays should be also considered in order to answer the following questions: i) what are the entrapment efficiency of resveratrol in the lipid based nanocarriers prepared in this work? ii) Resveratrol is preferentially entrapped in the water phase or in the lipid phase, as previously reveled by other polyphenols - Journal of Functional Foods, 46: 335-344. Doi: 10.1016/j.jff.2018.05.009); iii) How the concentration of the lipids that support the bilayer affect the entrapment efficiency of resveratrol? IV) What is the role of the structural differences of the mitochondriotropic lipid derivative (e.g. TPP-DSPE-PEG, DQA-DSPE-PEG) on the resveratrol entrapment efficiency and liposomes stability?

4.     In the section “In vitro cellular uptake and mitochondrial targeting” are described the results obtained with a procedure that allow to obtain a mitochondria-enriched fraction not highly purified mitochondria, so the measured resveratrol amounts can be located in mitochondria and in other sub-cellular fractions obtained with mitochondria. This limitation of the methodology should be considered considering the different possible mechanism of cellular uptake possible for nano-liposomes.

5.     The increase of ROS generation and mitochondrial membrane depolarization, both detected upon cell treatment with TLS(Res)- or DLS(Res)-liposomes are relevant indicators to characterize their cell effects. However, they cannot be used to state that liposomes are able to induce cell death by apoptosis (which can be proved combining caspase assays with the detection of typical morphological changes in cells). Additionally, the evaluation of the effects on non-cancer cells are also important to evaluate the selectivity of TLS(Res)- or DLS(Res)-liposomes for cancer cells, and characterize  their potential as therapeutic tool for cancer. Thus, additional assays with non-cancer cells should be included.

Author Response

Response to Reviewer 2 Comments

Manuscript ID: pharmaceutics-515761

Title: “Enhanced subcellular trafficking of resveratrol using mitochondriotropic liposomes in cancer cells”

The manuscript “Enhanced subcellular trafficking of resveratrol using mitochondriotropic liposomes in cancer cells” is a research paper that address the use a lipid-based nanocarriers to target resveratrol for mitochondria in order to induce apoptosis in cancer cells. The lipid-based nanocarriers are liposomes containing one mitochondriotropic cationic lipid-derivative, namely: Carboxybutyl triphenylphosphonium bromide-polyethylene glycol-distearoylphosphatidylethanolamine (TPP-DSPE-PEG) or dequalinium-polyethylene glycol distearoylphosphatidylethanolamine (DQA-DSPE-PEG).

1.       The mitochondriotropic lipid-derivatives were synthetized considering traditional procedures, and authors use NMR and MALDI-TOF mass to prove that the synthesis of TPP-DSPE-PEG and DQA-DSPE-PEG conjugates was well-succeeded, as shown in figure 3. However, the procedures for NMR and MALDI-TOF mass were not included in the Materials and Methods section. This procedures are important and should be included in the new version of manuscript.

2.        

Author’s response:

As suggested, the procedures were added in the method section as followed:

“and the formed product was characterized by proton NMR spectroscopy (Brucker, 600 MHz, Billerica, MA) and MALDI-TOF mass spectrometer (AXIMA-Assurance, Shimadzu, Kyoto, Japan).

2.     The subsection “2.3. Preparation of TPP-DSPE-PEG-modified liposomes carrying resveratrol (TLS(Res)) and DQA-DSPE-PEG-modified liposomes carrying resveratrol (DLS(Res)” must be rewritten in order to remove the errors related with the concentration units of each component [e.g. The following lipids (total amount of 2 mg) were dissolved in chloroform: POPC (final concentration = 7 μmol) … ]. The amount the components is confused with concentration. The liposomes composition should be indicated considering the molar proportions among all components (e.g. - POPC:Cholesterol: PEG-PE: resveratrol; 7/3/0.15/0.438, in μmoles), which allow to calculate the molar ratio of each one. The final volume of water solution buffer used to prepare the liposomes should be indicated.

Author’s response:

As suggested, the ratio was rephrased and final volume of buffer was added in the method section as followed:

“The following lipids (total amount of 2 mg) were dissolved in chloroform: POPC:cholesterol:PEG-PE (7:3:0.15 in μmol) and 100 μg of resveratrol dissolved in ethanol was added to the lipid/chloroform solution.”

“The organic solvents (chloroform and ethanol) were removed by vacuum evaporation, then the dried lipid film was mixed with 1 mL HBG buffer (10 mM HEPES, 5 % glucose, pH 7.4) and incubated at room temperature for 4 h with intermittent shaking.”

3.     The mitochondriotropic liposomes were characterized, considering the size and surface charge, assessed in terms of zeta potential. However, several other properties (and aspects) must be considered to show their potential as a therapeutic tool. Thus, a rationalization of the use of above-mentioned molar proportions of each lipid component (and cationic mitochondriotropic lipid derivative) will be also included in the new version of the manuscript;

Author’s response:

As per the comments, a rationalization of the use of molar proportions of each lipid component was included in the results section as followed:

“In this study, the mitochondriotropic liposomes were constructed with rational lipid composition and thin-film hydration followed by extrusion method according to previously published method [34,44], the amount of mitochondrial targeting ligand was decided as amount to exert a targeting effect [2,45,46].”

and additional assays should be also considered in order to answer the following questions: i) what are the entrapment efficiency of resveratrol in the lipid based nanocarriers prepared in this work?

Author’s response:

In this study, we didn’t calculate the entrapment efficiency, but we can expect that the entrapment efficiency of resveratrol in the liposoems would be more than 95% based on our previous publication (Colloids Surf B Biointerfaces 2015, Ko et al.).

 ii) Resveratrol is preferentially entrapped in the water phase or in the lipid phase, as previously reveled by other polyphenols - Journal of Functional Foods, 46: 335-344. Doi: 10.1016/j.jff.2018.05.009); 

Author’s response:

Resveratrol is relatively hydrophobic because of its planar stilbene motif (British Journal of Pharmacology, 2007, Xia et al.,), then the hydrophobic nature of resveratrol considerably contributes to its poor water solubility (Biomedicine, 2018, Salehi et al.). In case of hydrophobic drug, it resides in the acyl hydrocarbon chain of the liposome, thus resveratrol is preferentially entrapped in the lipid phase.

iii) How the concentration of the lipids that support the bilayer affect the entrapment efficiency of resveratrol?

Author’s response:

The high concentration of the lipids will lead to high entrapment efficiency of resveratrol, because the insertion of resveratrol into liposomes occurs due to hydrophobic interactions and association with phosphatidylcholine bilayer structures.

IV) What is the role of the structural differences of the mitochondriotropic lipid derivative (e.g. TPP-DSPE-PEG, DQA-DSPE-PEG) on the resveratrol entrapment efficiency and liposomes stability?

Author’s response:

The mitochondriotropic lipid derivative will affect only the mitochondrial targeting ability of the liposomes, not the resveratrol entrapment efficiency and liposomes stability, because that of only 1.5% per whole lipids was located on the surface.

4.     In the section “In vitro cellular uptake and mitochondrial targeting” are described the results obtained with a procedure that allow to obtain a mitochondria-enriched fraction not highly purified mitochondria, so the measured resveratrol amounts can be located in mitochondria and in other sub-cellular fractions obtained with mitochondria. This limitation of the methodology should be considered considering the different possible mechanism of cellular uptake possible for nano-liposomes.

Author’s response:

Although the mitochondria-enriched fraction contained not only mitochondria but also sub-cellular, it can be expected that the most of contents in that would be mitochondria. Therefore, the experimental procedure would support to evaluate the comparable mitochondrial accumulation of resveratrol delivered by mitochondriotropic liposomes.

5.     The increase of ROS generation and mitochondrial membrane depolarization, both detected upon cell treatment with TLS(Res)- or DLS(Res)-liposomes are relevant indicators to characterize their cell effects. However, they cannot be used to state that liposomes are able to induce cell death by apoptosis (which can be proved combining caspase assays with the detection of typical morphological changes in cells).

Author’s response:

It was reported that ROS production can be triggered by mitochondria dysfunction and cause damage to cells, which can lead to activation of cell death processes such as apoptosis (BBA-Molecular cell research, 2016, Redza-Dutordoor et al., Biomed Res Int, 2015, Kamogashira et al., Biochimie, 2002, Fleury et al.). Although the caspase assay was not performed in this study, therefore, it can be expected that the higher ROS generation and mitochondrial membrane depolarization of TLS(Res) and DLS(Res) induced the mitochondria dysfunction, which can lead to apoptosis and necrosis of cells.

Additionally, the evaluation of the effects on non-cancer cells are also important to evaluate the selectivity of TLS(Res)- or DLS(Res)-liposomes for cancer cells, and characterize  their potential as therapeutic tool for cancer. Thus, additional assays with non-cancer cells should be included.

Author’s response:

As commented, the evaluation of the effects on non-cancer cells may contrast the selectivity of TLS(Res) or DLS(Res) for cancer cells and their potential as therapeutic tool for cancer. In this study, however, we mainly focused on the selectivity of mitochondriotropic liposomes for the mitochondria, not that for the cancer cells. Considering the structure of mitochondriotropic liposomes, which have only mitochondrial-targeting moiety without any cancer-targeting moiety on the surface, it could be reasonably assumed that the liposomes in non-cancer cells show very similar in vitro results in cancer cells.

Reviewer 3 Report

This manuscript entitled "Enhanced subcellular trafficking of resveratrol using  mitochondriotropic liposomes in cancer cells" showed that the mitochondria-targeting liposomes were successfully synthesized. The TLS(Res) and DLS(Res) accumulated significantly into the mitochondria. These data comprise a valuable contribution to treat cancers by mitochondrial targeting delivery of therapeutics. However, based on the results, I have a few comments below that I would like see addressed.

Q1: The resolution of Figure 6 is too low to see the mitochondria. It is recommended to provide an enlarged mitochondrial picture so that the accumulation of the drug can be clearly seen.

Q2: There is no S.E.M in the "whole column chart" of Figure 7, please add.

Q3: How many times experiment have been repeated in Figures 8 and 9? Is it the same every time? Quantitative statistics should be given.

Author Response

This manuscript entitled "Enhanced subcellular trafficking of resveratrol using mitochondriotropic liposomes in cancer cells" showed that the mitochondria-targeting liposomes were successfully synthesized. The TLS(Res) and DLS(Res) accumulated significantly into the mitochondria. These data comprise a valuable contribution to treat cancers by mitochondrial targeting delivery of therapeutics. However, based on the results, I have a few comments below that I would like see addressed.

Q1: The resolution of Figure 6 is too low to see the mitochondria. It is recommended to provide an enlarged mitochondrial picture so that the accumulation of the drug can be clearly seen.

Author’s response:

Please be noted that images of high quality were uploaded in separate files.

Q2: There is no S.E.M in the "whole column chart" of Figure 7, please add.

Author’s response:

In the Figure 7, each fractional resveratrol amount was calculated as percentage of the whole cellular uptake amount of resveratrol. Therefore, each whole mean value is 100% with S.E.M of zero.

Q3: How many times experiment have been repeated in Figures 8 and 9? Is it the same every time? Quantitative statistics should be given.

Author’s response:

The experiments related with Figure 8 and 9 have been repeated as 3 times. Please be note that the description of quantitative statistics in Figure 8 was presented in the result part as followed:

The mean of fluorescence intensity in each group was calculated as TLS(Res) (1024.5 ± 12.3), DLS(Res) (1799.1 ± 14.9), LS(Res) (842.2 ± 7.9), free resveratrol (705.3 ± 17.1) and positive control (619.3 ± 25.1).[Page 7 line 294 – 296]

Additionally, the description of quantitative statistics in Figure 9 was presented in the result part as followed:

As shown in Figure 9, the percentage of mitochondrial membrane depolarization was 7.95 ± 1.32 %, 13.41 ± 2.95 %, 24.78 ± 4.19 %, and 32.45 ± 3.07 % in the free resveratrol, LS(Res), TLS(Res) and DLS(Res)-treated cells, respectively.” [Page 7 line 307 – 309]

Round 2

Reviewer 1 Report

In the revised version the authors made no effort to improve their manuscript and their responses are far from adequate. In brief, the authors did not perform any cell toxicity experiments (which is fundamental for the usefulness or not of their systems), did not examine if apoptosis (as claimed in the abstract) or necrosis is induced, and control experiments are still missing, namely the effect of empty TLS and DLS. They performed no proton NMR analysis, no 13C NMR experiments (simply by including, only in the revised manuscript, of a  purification step does not provide proof that purification is indeed attained) and no effort to provide better/detailed confocal images of cells treated with their systems. Finally they insist that liposomes remain intact until they penetrate the mitochondrion (Fig. 1) without providing any proof. The references they provide as an answer, are their references and the first one that I read  (Ko, Y.T.; Bhattacharya, R.; Bickel, U. Liposome encapsulated polyethylenimine/457 ODN polyplexes for brain targeting. Journal of controlled release 2009) was not at all relevant to this point.

Author Response

In the revised version the authors made no effort to improve their manuscript and their responses are far from adequate. In brief, the authors did not perform any cell toxicity experiments (which is fundamental for the usefulness or not of their systems), did not examine if apoptosis (as claimed in the abstract) or necrosis is induced, and control experiments are still missing, namely the effect of empty TLS and DLS.

Authors’s response:

As pointed out, the cell toxicity experiments with empty mitochondriotropic liposomes and drug-loaded mitochondriotropic liposomes were carried out and then the manuscript was revised by adding the method, figure, results and discussion and modifying the abstract and conclusion as followed:

“Furthermore, TLS(Res) and DLS(Res) induced cytotoxicity of cancer cells by generating reactive oxygen species (ROS) and by dissipating the mitochondrial membrane potential.” [Page 1 line 21 – 23]

2.6 In vitro cytotoxicity assay

The cytotoxicity of the LS, TLS, DLS, free Res, LS(Res), TLS(Res) and DLS(Res) was assessed using the MTT assay. Briefly, B16F10 were seeded in 96-well plates at a density of 1 × 104 cells/well and incubated overnight at optimal condition. The cells were treated by replacing medium with fresh serum-free medium containing a range of concentrations of LS, TLS, DLS, free Res, LS(Res), TLS(Res) and DLS(Res). Following incubation for 24 h at 37 °C, the cells were washed twice with PBS, then incubated with serum-free medium containing water-soluble tetrazolium (WST) solution (Ez-Cytox Cell Viability Assay Kit, DoGen, South Korea) for 30 min at 37 °C in the dark. The absorbance was measured at 480 nm using a microplate reader (Epoch, BioTek Instruments, Winooski, VT).” [Page 3 line 147 – 155]

“3.3 In vitro cellular cytotoxicity

Figure 6A shows the cytotoxicity of the non-targeting (LS) and mitochondria-targeting liposomes (TLS and DLS) without resveratrol in B16F10 cells. Cell viability showed no significant differences among cells treated with LS, TLS and DLS for 24 h. The average cell viability showed a higher than 90% up to the liposomes concentration of 500 μg/mL. This result indicated that mitochondriotropic liposomes did not seriously damage to cells, demonstrating that the liposomes itself did not contribute to the cellular toxicity.

To further evaluate the cytotoxicity of the mitochondria-targeting liposomes carrying resveratrol, MTT assay were carried out with different concentration of the liposomes. As shown Figure 6B, the free Res, LS(Res), TLS(Res) and DLS(Res) showed dose-dependent cytotoxicity against B16F10 cells. In particular, the mitochondriotropic liposomes carrying resveratrol (TLS(Res) and DLS(Res)) showed the lower viability than free Res and LS(Res), indicating that the mitochondria-targeting liposomes improves the cytotoxicity efficacy of resveratrol in cancer cells.” [Page 6 line 271 – 283]

“TLS(Res) and DLS(Res) showed superior in vitro behavior in the B16F10 cells, including increased accumulation in mitochondria, anticancer efficacy, ROS generation, and mitochondrial depolarization, compared to that of LS(Res).” [Page 8 line 348 – 350]

“Figure 6. In vitro viability of B16F10 cells after applying the liposomes (A) non-carrying resveratrol and (B) carrying resveratrol at 24 h. (Mean ± S.E.M, n=4). * indicates a statistical difference from the LS(Res), p<0.05.” [Page 16 line 547 – 551]

Additionally, please be noted that the order of the methods and results section and figures was rearranged by adding the cytotoxicity data.

They performed no proton NMR analysis, no 13C NMR experiments (simply by including, only in the revised manuscript, of a purification step does not provide proof that purification is indeed attained) and no effort to provide better/detailed confocal images of cells treated with their systems.

Authors’s response:

Although 13C NMR experiment and data and detailed confocal images of cells treated with our systems commented by the reviewer would obviously make our study more complete and comprehensive, we believe that the current status of data sets would be sufficient to demonstrate the specific aim that our delivery systems could provide a potential strategy to treat cancers by subcellular targeting delivery of therapeutics and stimulation of the mitochondrial signaling pathway.

Finally they insist that liposomes remain intact until they penetrate the mitochondrion (Fig. 1) without providing any proof. The references they provide as an answer, are their references and the first one that I read (Ko, Y.T.; Bhattacharya, R.; Bickel, U. Liposome encapsulated polyethylenimine/457 ODN polyplexes for brain targeting. Journal of controlled release 2009) was not at all relevant to this point.

Authors’s response:

At first, we apologize for providing non-relevant reference.

Based on the studies related with successful liposomal delivery for the subcellular trafficking through endosomal escape, we can expect that our liposomal system was also well established for the endosomal escape. To further support our claim, we provide additional references (JCR 2012 Swati Biswas, et al., Biomaterials 2011 Xiao-Xing Wang, et al., Nanomedicine(lond) 2015 Alessandro Parodi, et al.)

Reviewer 2 Report

The paper was improved, which is particularly evidenced in the methods section of the new version of the manuscript. However, the follow concerns, raised in the first review, was not adequately considered in the new version of the manuscript, despite some were well answered in the letter to the reviewer.

In relation to the question (in my comment 3) i) what are the entrapment efficiency of resveratrol in the lipid based nanocarriers prepared in this work?

Author’s response: In this study, we didn’t calculate the entrapment efficiency, but we can expect that the entrapment efficiency of resveratrol in the liposoems would be more than 95% based on our previous publication (Colloids Surf B Biointerfaces 2015, Ko et al.). But in the manuscript the following sentence is written (lines 250,251), “The encapsulation efficiencies of resveratrol in liposomal carriers were found to be more than 90 % (data not shown).” Thus,“Data not shown” should be replaced by (Colloids Surf B Biointerfaces 2015, Ko et al.).

In relation to my previous comment 5: i) the increase of ROS generation and mitochondrial membrane depolarization (evaluated in paper) are important pathological features that occurs during apoptosis running, but by themselves do not guarantee that cell death by apoptosis. Apoptosis is revealed by caspase activation and by the histopathological
markers of nuclear DNA. These limitations should be incorporate in the manuscript, results section.

ii) the evaluation of the effects on non-cancer cells are also important to evaluate the selectivity of TLS(Res)- or DLS(Res)-liposomes for cancer cells, and characterizetheir potential as therapeutic tool for cancer.

Author’s response: “As commented, the evaluation of the effects on non-cancer cells may contrast the selectivity of TLS(Res) or DLS(Res) for cancer cells and their potential as therapeutic tool for cancer. In this study, however, we mainly focused on the selectivity of mitochondriotropic liposomes for the mitochondria, not that for the cancer cells. Considering the structure of mitochondriotropic liposomes, which have only mitochondrial-targeting moiety without any
cancer-targeting moiety on the surface, it could be reasonably assumed that the liposomes in non-cancer cells show very similar in vitro results in cancer cells.”

The above-underlined sentence indicates that the authors recognize the lack of selectivity of these liposomes for cancer cells, which can be minimized by the addition of a cancer-targeting moiety on its surface. But it was not assessed nor included in the manuscript. This limitation can be included in manuscript, for instance, modifying the last sentence
in the conclusions for something like: Since the selectivity of liposomes for cancer cells can be improved by adding a cancer-targeting moiety on their surface, the mitochondria-targeting liposomes carrying resveratrol provide a potential strategy for cancer treatment by mitochondrial targeting delivery of therapeutics and stimulation the mitochondrial signaling pathway.

Author Response

The paper was improved, which is particularly evidenced in the methods section of the new version of the manuscript. However, the follow concerns, raised in the first review, was not adequately considered in the new version of the manuscript, despite some were well answered in the letter to the reviewer.

In relation to the question (in my comment 3) i) what are the entrapment efficiency of resveratrol in the lipid based nanocarriers prepared in this work?

Author’s response: In this study, we didn’t calculate the entrapment efficiency, but we can expect that the entrapment efficiency of resveratrol in the liposoems would be more than 95% based on our previous publication (Colloids Surf B Biointerfaces 2015, Ko et al.). But in the manuscript the following sentence is written (lines 250, 251), “The encapsulation efficiencies of resveratrol in liposomal carriers were found to be more than 90 % (data not shown).” Thus,“Data not shown” should be replaced by (Colloids Surf B Biointerfaces 2015, Ko et al.).

Author’s response:

As pointed out, the “data not shown” was replaced by reference no. 46 (Colloids Surf B Biointerfaces 2015, Ko et al.) and the sentence was revised to provide further details of loading capacity of extruded liposomes by adding more references as followed:

“It was reported that the loading capacity of extruded liposomes carrying resveratrol showed more than 5 % and the encapsulation efficiencies of resveratrol in liposomal carriers were found to be more than 90 % [2, 46-48]” [Page 6 line 259 – 261]

In relation to my previous comment 5: i) the increase of ROS generation and mitochondrial membrane depolarization (evaluated in paper) are important pathological features that occurs during apoptosis running, but by themselves do not guarantee that cell death by apoptosis. Apoptosis is revealed by caspase activation and by the histopathological markers of nuclear DNA. These limitations should be incorporate in the manuscript, results section.

ii) the evaluation of the effects on non-cancer cells are also important to evaluate the selectivity of TLS(Res)- or DLS(Res)-liposomes for cancer cells, and characterize their potential as therapeutic tool for cancer.

Author’s response: “As commented, the evaluation of the effects on non-cancer cells may contrast the selectivity of TLS(Res) or DLS(Res) for cancer cells and their potential as therapeutic tool for cancer. In this study, however, we mainly focused on the selectivity of mitochondriotropic liposomes for the mitochondria, not that for the cancer cells. Considering the structure of mitochondriotropic liposomes, which have only mitochondrial-targeting moiety without any cancer-targeting moiety on the surface, it could be reasonably assumed that the liposomes in non-cancer cells show very similar in vitro results in cancer cells.”

The above-underlined sentence indicates that the authors recognize the lack of selectivity of these liposomes for cancer cells, which can be minimized by the addition of a cancer-targeting moiety on its surface. But it was not assessed nor included in the manuscript. This limitation can be included in manuscript, for instance, modifying the last sentence in the conclusions for something like: Since the selectivity of liposomes for cancer cells can be improved by adding a cancer-targeting moiety on their surface, the mitochondria-targeting liposomes carrying resveratrol provide a potential strategy for cancer treatment by mitochondrial targeting delivery of therapeutics and stimulation the mitochondrial signaling pathway.

Author’s response:

As suggested, the last sentence in the conclusion was modified as followed:

“Since the selectivity of liposomes for cancer cells can be improved by adding a cancer-targeting moiety on their surface, the mitochondria-targeting liposomes carrying resveratrol provide a potential strategy for cancer treatment by mitochondrial targeting delivery of therapeutics and stimulation the mitochondrial signaling pathway.” [Page 8 line 350 – 353]
